# Membrane Targeting and GTPase Activity of Rab7 Are Required for Its Ubiquitination by RNF167

**DOI:** 10.3390/ijms23147847

**Published:** 2022-07-16

**Authors:** Kim Ghilarducci, Valérie C. Cabana, Ali Harake, Laurent Cappadocia, Marc P. Lussier

**Affiliations:** 1Département de Chimie, Université du Québec à Montréal, Montréal, QC H2X 2J6, Canada; ghilarducci.kim@courrier.uqam.ca (K.G.); cabana.valerie@courrier.uqam.ca (V.C.C.); harake.ali@courrier.uqam.ca (A.H.); cappadocia.laurent@uqam.ca (L.C.); 2Centre d’Excellence en Recherche sur les Maladies Orphelines-Fondation Courtois (CERMO-FC), Université du Québec à Montréal, Montréal, QC H2X 3Y7, Canada

**Keywords:** RNF167, Rab7, ubiquitination, endosome, Charcot–Marie–Tooth Type 2B Rab7 variant

## Abstract

Rab7 is a GTPase that controls late endosome and lysosome trafficking. Recent studies have demonstrated that Rab7 is ubiquitinated, a post-translational modification mediated by an enzymatic cascade. To date, only one ubiquitin E3 ligase and one deubiquitinase have been identified in regulating Rab7 ubiquitination. Here, we report that RNF167, a transmembrane endolysosomal ubiquitin ligase, can ubiquitinate Rab7. Using immunoprecipitation and in vitro ubiquitination assays, we demonstrate that Rab7 is a direct substrate of RNF167. Subcellular fractionation indicates that RNF167 activity maintains Rab7′s membrane localization. Epifluorescence microscopy in HeLa cells shows that Rab7-positive vesicles are larger under conditions enabling Rab7 ubiquitination by RNF167. Characterization of its ubiquitination reveals that Rab7 must be in its GTP-bound active form for membrane anchoring and, thus, accessible for RNF167-mediated ubiquitin attachment. Cellular distribution analyses of lysosome marker Lamp1 show that vesicle positioning is independent of Rab7 and RNF167 expression and that Rab7 endosomal localization is not affected by RNF167 knockdown. However, both Rab7 and RNF167 depletion affect each other’s lysosomal localization. Finally, this study demonstrates that the RNF167-mediated ubiquitination of Rab7 GTPase is impaired by variants of Charcot–Marie–Tooth Type 2B disease. This study identified RNF167 as a new ubiquitin ligase for Rab7 while expanding our knowledge of the mechanisms underlying the ubiquitination of Rab7.

## 1. Introduction

Maintaining cellular homeostasis through appropriate membrane trafficking is biologically important and must be tightly regulated to avoid cell dysfunction [1]. The Rab proteins family of GTPases is an important group of enzymes that control membrane and vesicle trafficking throughout the cell [2]. These enzymes cycle through guanosine nucleotide-associated states, being active in the GTP-bound state and inactive when bound to GDP. To be activated, Rab proteins must be prenylated on cysteines in their C-terminal tails. The lipid modification allows Rab proteins to be anchored in membranes [3,4,5]. Once anchored at the appropriate vesicle membrane, guanine nucleotide exchange factors (GEFs) activate the Rab GTPase by exchanging the bound GDP for GTP [2]. Once activated, GTP-bound Rab can recruit effectors to induce one of its multiple functions in membrane tethering, trafficking or fusion and fission events [4]. GTPase-activating proteins (GAPs) stimulate GTP hydrolysis returning the Rab protein to its GDP-bound, inactive state [2]. Finally, the GDP dissociation inhibitor (GDI) removes Rab from the membrane to stabilize inactive protein in the cytosol. The activation cycle of the GTPase can start once the GDI returns the protein to the membrane [6].

Different GTPases are involved in endosome and lysosome trafficking [4]. For example, Rabs 4 and 5 are involved in early endosome trafficking [7,8,9,10], whereas Rabs 11 and 25 are implicated in recycling endosome trafficking [11,12]. Late endosome to trans Golgi network (TGN) trafficking and late endosome and lysosome trafficking are regulated by Rab7A (named Rab7 hereafter) and Rab9 proteins [13,14,15,16]. The *RAB7A* gene encodes a protein that is found in eukaryotes throughout evolution [17]; recent studies have demonstrated that Rab7 protein regulation can be monitored by post-translational modifications, such as phosphorylation and ubiquitination [18,19,20,21,22].

Protein ubiquitination is characterized by the covalent attachment of ubiquitin (UB), a small protein of 76 amino acids, typically to lysine residues [23,24]. This modification is mediated by an enzymatic cascade involving a UB activation enzyme E1 (UBE1), a UB conjugating enzyme E2 (UBE2) and a UB ligase (UBE3) [25]. First, UBE1 uses ATP to activate and subsequently bind UB to the UBE1 active cysteine via the diglycine C-terminus of UB [26]. Then, UBE1 transfers UB to an active cysteine of UBE2 [24]. Finally, UBE2 conjugated to UB interacts with UBE3 to either (i) transfer UB to a substrate or (ii) transfer UB to an active UBE3 cysteine for subsequent UBE3-mediated UB transfer to a substrate. In mono-ubiquitination, a single UB moiety is attached via its C-terminal glycine to a specific substrate lysine. Mono-ubiquitination is common and is involved in DNA repair, endocytosis, and vesicular trafficking [27,28], and serves as a probe for UBE2-dependant specific UB chains such as the K63 chain by the UBE2N–UBE2V1/V2 complex [29] or the K11 chain by UBE2S [30]. If necessary, additional UB moieties can be attached to the preceding UB molecule, forming an elongated chain in a process called poly-ubiquitination. Ultimately, the functional outcomes of substrate ubiquitination in the cell are governed by the UB code, which typically relies on the specific type of UB chain. For instance, the most common UB chain using K48 residues is associated with proteasome-mediated degradation, whereas the K63-linked UB chain is involved in DNA repair, membrane protein endocytosis and lysosomal degradation [31]. Importantly, UB modification is reversible by the action of deubiquitinase enzymes [32].

Major ubiquitination sites of Rab7 protein include K38, K191 and K194 [20,21,22]. K38 ubiquitination of Rab7 protein is mediated by the UB ligase (UBE3) Parkin. It has been shown that this ubiquitination is involved in the binding of Rab7 with its partner Rab-interacting lysosomal protein (RILP) [20]. Interaction with RILP leads to the recruitment of the homotypic fusion and protein sorting (HOPS) complex to induce early endosome and late endosome fusion and late endosome and lysosome fusion [33]. Rab7 recruitment of RILP is also involved in dynein interaction, leading to retrograde transport on microtubules [34]. Single lysine to arginine residue substitution of K191 or K194 within Rab7 protein leads to the same phenotype in the cell [20,22], suggesting that both ubiquitination sites can affect Rab7 function similarly. However, divergence exists in the literature regarding how those two ubiquitination sites are involved in Rab7 membrane association. One study, that simultaneously mutated K191 and K194, indicated that K191 ubiquitination leads to membrane tubulation, demonstrated by an increase in retromer complex recruitment when Rab7 is ubiquitinated [21], which induces retrograde transport between late endosomes and the TGN [35]. The study concluded that Rab7 ubiquitination retained the GTPase on the membrane, whereas its removal by the deubiquitinase USP32 was important for Rab7 to leave the membrane [21]. Another study, using computational models where both lysines were simultaneously ubiquitinated, indicated that K191 and K194 ubiquitination prevents Rab7 membrane association [22]. Although a deubiquitinase was found for the K191 ubiquitination site, no UBE3 was shown to be specific for either the K191 or K194 site.

To date, there is only one Rab7-specific UBE3 ligase. In this study, we sought to characterize a novel UB ligase for Rab7. Specifically, we show that RNF167, an endolysosomal transmembrane RING-domain containing UBE3 ligase, is involved in Rab7 K191 or K194 ubiquitination. The *RNF167* gene encodes a protein homologue of the Godzilla UBE3 ligase in *Drosophila melanogaster* [36,37] and is conserved in many organisms, including humans, rats, zebrafish, and *Caenorhabditis elegans*. Immunofluorescence assays in HeLa cells indicate that Rab7- and RNF167-positive vesicles are larger in situations that promote Rab7 ubiquitination. We also demonstrate in HEK293T cells that Rab7 activation and membrane association are required for RNF167-mediated ubiquitination. Rab7 is also required for RNF167-dependant Lamp1-positive vesicle perinuclear distribution in HeLa cells, but Rab7 does not require RNF167 for vesicle distribution. Finally, RNF167-mediated ubiquitination of Rab7 GTPase domain variants from Charcot–Marie–Tooth Type 2B (CMT2B) disease is impaired.

## 2. Results

### 2.1. Rab7 Protein Is a Substrate of RNF167

Protein ubiquitination of Rab GTPases, such as Rabs 5, 7 and 11, regulates endosomal vesicles and receptor trafficking [20,21,38,39]. Although the ubiquitination of Rab7 is associated with Parkin activity [20], Rab7 and 9 proteins colocalize with the endosomal ubiquitin ligase RNF167 [40]. Due to its presence at the plasma membrane and in endolysosomes [40], RNF167 is expected to traffic through many Rab-positive endosomal compartments. To explore whether Rab GTPases involved in endolysosomal trafficking could be a substrate of RNF167, we performed an in vivo ubiquitination assay. Plasmids encoding the green fluorescent protein (GFP) fused to Rabs 5, 7, 9 and 11 were separately transfected into HEK293T cells with either a control vector or plasmid encoding RNF167-V5-His. After cell lysis, GFP-Rab proteins were immunopurified (IP), and specific immunoblotting against UB revealed that exogenous expression of RNF167 mediates Rab7 and Rab9 ubiquitination (Figure 1A). Rabs 5 and 11 were not ubiquitinated by RNF167 under our experimental conditions.

RNF167 activity is associated with late endosome and lysosome retrograde transport [41]; therefore, we decided to further characterize Rab7 ubiquitination. First, to confirm that Rab7 ubiquitination depends on RNF167 activity, the co-transfection of plasmids encoding GFP-Rab7 with either RNF167 wild type (WT) or its dominant-negative (DN) variant H250W/H253W [40,42] showed that WT, but not DN, RNF167 led to Rab7 ubiquitination (Figure 1B). This result supports the hypothesis that RNF167 acts as a UB ligase for Rab7. Then, because RNF167 must physically associate with Rab7 to promote its ubiquitination, we tested whether Rab7 and RNF167 interact. Similarly to results reported for another RNF167 substrate [42], our experiments showed that a higher amount of DN RNF167, compared with WT RNF167, co-immunoprecipitated with Rab7 (Figure 1C). This result suggests that the ability of Rab7 to bind RNF167 is independent of RING domain activity and that the absence of Rab7 ubiquitination in the presence of DN RNF167 is not due to a reduced binding capability (Figure 1B). Finally, to corroborate that RNF167 mediates Rab7 ubiquitination, we conducted an in vitro ubiquitination assay composed of purified HA-RNF167-6xHis (a.a. 194–350), purified GST-Rab7, ATP, UB, UBE1, and the highly reactive UBE2D1 [24] that functionally interacts with RNF167 [42,43]. Using GST-Rab7 pull-downs and immunoblotting against UB, we observed Rab7 ubiquitination when RNF167 and UBE2D1 activities were unaltered (Figure 1D). Together, these data indicate that Rab7 interacts with and is efficiently ubiquitinated by RNF167.

### 2.2. RNF167 Promotes Rab7 Membrane Localization

Recent studies identified three major sites of ubiquitination within the Rab7 protein [20,21,22]. Thus, we tested the location of RNF167-mediated Rab7 protein ubiquitination in HEK293T cells by co-transfecting plasmids encoding RNF167 with either GFP-Rab7 WT, K38R or 2KR, which combines the lysine-to-arginine substitution of both K191 and K194, as per prior studies performed by others [20,21,22]. Immunoblot of UB-modified Rab7 was performed after IP of GFP-Rab7 from cell lysates. The results showed that, unlike WT and K38R versions, the Rab7 2KR variant was not ubiquitinated, indicating the importance of these residues to RNF167-mediated Rab7 ubiquitination (Figure 2A).

To determine whether Rab7 protein ubiquitination influences its cellular localization, we tested Rab7 membrane association in the presence of WT or DN RNF167. Membrane and cytosol fractions were prepared from HEK293T cells transfected with a plasmid encoding the 2xHA-Rab7 protein in combination with either control vector or plasmid encoding WT or DN RNF167-V5-His. Here, 2xHA-Rab7 was used to minimize the possible influence of a large high molecular weight tag (i.e., such as GFP) on Rab7 protein localization and ubiquitination. Proteins from the collected fractions were separated onto SDS-PAGE before specific immunoblots were performed. The results revealed the quality of the cytosolic and membrane fractionations as per the immunoblots for the endogenous glyceraldehyde-3-phosphate dehydrogenase (GAPDH) cytosolic protein and the transferrin receptor membrane protein (Figure 2B). As expected, Rab7 protein was found in both cytosolic and membrane fractions. The Rab7 membrane-to-cytosol ratio was analyzed using the stain-free signal as a normalization standard. The results indicate that WT, but not DN, RNF167 significantly increases Rab7 membrane association (WT, 1.422 ± 0.046, *p* = 0.0459, DN, 1.095 ± 0.060, *p* > 0.9999), suggesting that Rab7 ubiquitination by RNF167 promotes Rab7 association with membranes (Figure 2B).

The protein ubiquitination of Rab7 at K191 increases vesicle diameter [21]; therefore, plasmid encoding WT 2xHA-Rab7 was transfected in combination with either a control vector or plasmids encoding WT or DN RNF167 in HeLa cells. Using immunofluorescence assays (Figure 2C,D), Rab7-positive vesicles in cells expressing WT RNF167 are larger than the control condition (CTL, 1.306 ± 0.037 µm, WT, 1.442 ± 0.025 µm, *p* = 0.0177). In contrast, the expression of DN RNF167 leads to smaller vesicles (DN, 1.078 ± 0.025 µm, *p* < 0.0001). Vesicle enlargement does not reflect higher RNF167 and Rab7 colocalization, because the Manders’ overlap coefficient for Rab7 with WT RNF167 is significantly lower than with DN RNF167 (Rab7-WT RNF167, 0.212 ± 0.018, Rab7-DN RNF167, 0.289 ± 0.018, *p* = 0.0025) (Figure 2E). To further investigate the impact of RNF167-mediated ubiquitination on Rab7-positive vesicle size, we tested whether RNF167 activity modifies the size of WT, K38R or 2KR Rab7-positives vesicles (Figure 2F,G). The results show that Rab7 2KR vesicles are significantly smaller than WT-positive vesicles, whereas the diameter of Rab7 K38R vesicles is unchanged (WT, 0.992 ± 0.017 µm; 2KR, 0.747 ± 0.016 µm, *p* < 0.0001; K38R, 1.019 ± 0.016 µm, *p* = 0.8758). This suggests that Rab7 ubiquitination at K191 or K194 controls the Rab7-positive vesicle size. Concomitantly, RNF167-mediated Rab7 ubiquitination promotes larger Rab7 WT vesicles (WT + RNF167, 1.081 ± 0.018 µm, *p* = 0.0028), whereas RNF167 does not change the diameter of vesicles positive for either Rab7 2KR or Rab7 K38R (2KR + RNF167, 0.718 ± 0.017 µm, *p* = 0.7219; K38R + RNF167, 1.071 ± 0.017 µm, *p* = 0.2114) (Figure 2G). The Manders’ overlap coefficient of WT/K38R/2KR Rab7 with WT RNF167 was not significantly different (Rab7 WT/RNF167, 0.278 ± 0.022; Rab7 2KR/RNF167, 0.312 ± 0.026, *p* = 0.9811; Rab7 K38R/RNF167, 0.232 ± 0.019, *p* = 0.3040) (Figure 2H). Together, these results suggest that the RNF167-mediated ubiquitination of Rab7 increases membrane association and enlarges vesicles.

### 2.3. Rab7 Ubiquitination Depends on Its Active Conformation and Membrane Anchoring

Rab7 is known to switch between an inactive GDP-bound state and an active GTP-bound state that regulates its recruitment to endosomes [2]. Our results thus far suggest that the ubiquitination of Rab7 protein modulates its membrane association; therefore, we investigated whether Rab7’s active or inactive state influences its ubiquitination by RNF167. To this end, we performed a ubiquitination assay in HEK293T cells co-expressing WT RNF167 with either Rab7 WT, the constitutively activated Rab7 Q67L or the inactive dominant-negative Rab7 T22N mutant. The results showed that Rab7 Q67L is ubiquitinated by RNF167, with ubiquitination levels higher than WT Rab7 (Figure 3A), which contrasts with the inactive Rab7 T22N mutant that is not ubiquitinated by RNF167. Reproducibly, the ubiquitination of Rab7 Q67L was detected even in the absence of RNF167 exogenous expression (Figure 3A). These results suggest that Rab7 must be in an active state for its ubiquitination by RNF167. It remained unclear, however, whether Rab7 membrane association is part of the mechanism leading to its ubiquitination, or a consequence of being anchored at the membrane. C-terminal Rab7 amino acid residues C205 and C207 are known to be modified with geranyl–geranyl hydrophobic moieties that are required for Rab7 anchoring on endosomal membranes [44]. Therefore, to determine whether Rab7 ubiquitination mediated by RNF167 depends on membrane anchoring, HEK293T cells were transfected with a plasmid encoding either Rab7 WT or the C205S/C207S variant that cannot be anchored at the membrane [16]. After Rab7 IP and subsequent anti-UB immunoblot, the results showed that the Rab7 C205S/C207S mutant is not ubiquitinated by RNF167, suggesting that ubiquitination occurs after the membrane anchoring of Rab7 (Figure 3B). Together, these results indicate that Rab7 must be in an active state and anchored at the membrane for its RNF167-mediated ubiquitination.

### 2.4. Rab7 Controls RNF167 Positioning to the Lysosome

Recent studies have shown that RNF167 controls lysosome trafficking and positioning [41,45]. Rab7 is also involved in perinuclear lysosome positioning [13]; therefore, we investigated whether Rab7 could influence the RNF167-mediated positioning of lysosomes. To test this hypothesis, we used Dicer-Substrate Short Interfering RNAs (DsiRNAs) against human *RAB7A* in HeLa cells. Results from specific immunoblotting against endogenous Rab7 protein show that DsiRNA #1 and #2 were the most efficient at reducing Rab7 protein abundance (Figure 4A). Specific immunoblotting shows that reduction in the Rab7 protein level does not alter the total protein level of endogenous lysosomal marker Lamp1 (Figure 4B). Using immunofluorescence, we tested whether Rab7 is required for RNF167-mediated lysosome positioning by measuring the position of the Lamp1-lysosomal marker. Overall, the ratios of perinuclear to peripheral Lamp1-positive vesicles show that lysosome positioning is not convincingly affected in Rab7-depleted cells when compared with control cells or in the presence of exogenous RNF167 (siCTL, 2.681 ± 0.274; siRab7#1, 1.985 ± 0.229, *p* = 0.5781; siRab7#2, 1.686 ± 0.175, *p* = 0.1624; siCTL + RNF167, 3.639 ± 0.348, *p* = 0.1694; siRab7#1 + RNF167, 2.857 ± 0.385, *p* = 0.3571; siRab7#2 + RNF167, 1.684 ± 0.243, *p* = 0.0003) (Figure 4C,D). To test if the depletion of Rab7 impacts the localization of RNF167 to lysosomes, we determined the Manders’ overlap coefficients for RNF167 and Lamp1. The results show that Rab7 depletion decreases the overlap between RNF167 and Lamp1 (siCTL + RNF167, 0.312 ± 0.014; siRab7#1 + RNF167, 0.166 ± 0.014, *p* < 0.0001; siRab7#2 + RNF167, 0.190 ± 0.018, *p* < 0.0001) (Figure 4E), suggesting that Rab7 regulates RNF167 localization in Lamp1-positive vesicles.

### 2.5. RNF167 Protein Regulates Rab7 Localization to Lysosomes

Knowing that Rab7 abundance influences RNF167 lysosomal localization (Figure 4), we investigated whether reducing RNF167 levels impairs Rab7-associated lysosome distribution. First, we evaluated the efficiency of three DsiRNAs against *RNF167*. The results indicated that DsiRNA #1 and #2 against *RNF167* were the most efficient at reducing protein levels in HeLa cells (Figure 5A). Exogenous RNF167-V5-His was used here due to the limited success, at least in our hands, with antibodies probing endogenous RNF167 protein. Specific immunoblots of total cell lysates prepared from control and RNF167-depleted HeLa cells show that RNF167 depletion does not modify Lamp1 protein abundance (Figure 5B). Then, as shown in Figure 4, we analyzed the ratio of perinuclear to peripheral Lamp1-positive vesicles to test whether RNF167 is required for Rab7-dependent lysosome positioning in HeLa cells. The results indicated that the depletion of RNF167 does not importantly reduce the perinuclear to peripheral Lamp1 intensity ratio (siCTL, 2.790 ± 0.275; siRNF167#1, 1.490 ± 0.142, *p* < 0.0001; siRNF167#2, 2.295 ± 0.188, *p* = 0.4472). Similarly, the comparison of the exogenous level of Rab7 without or with RNF167 depletion does not significantly alter lysosome positioning (siCTL + Rab7, 1.990 ± 0.177; siRNF167#1 + Rab7, 2.064 ± 0.252, *p* = 0.9999; siRNF167#2 + Rab7, 1.674 ± 0.146, *p* = 0.8829) (Figure 5C,D). Together, this suggests that RNF167 does not influence the Rab7-mediated positioning of lysosomes in cells. Finally, to determine whether RNF167 impacts the lysosomal localization of Rab7, we calculated Manders’ overlap coefficient between Rab7 and Lamp1. The results indicate that the overlap between Rab7 and Lamp1 significantly increases in RNF167-depleted cells (siCTL + Rab7, 0.060 ± 0.005, siRNF167#1 + Rab7, 0.088 ± 0.007, *p* = 0.0018; siRNF167#2 + Rab7, 0.078 ± 0.006, *p* = 0.0334) (Figure 5E), thus suggesting that RNF167 activity normally reduces Rab7 localization in Lamp1-positive vesicles.

### 2.6. RNF167-Dependent Rab7 Ubiquitination Is Impaired in Charcot–Marie–Tooth Type 2B Variants

Charcot–Marie–Tooth Type 2 subtype B (CMT2B) disorder is due to genetic alteration in the *RAB7A* gene [46]. The genetic alterations generate protein variants surrounding the GTPase domain of Rab7 (Figure 6A). Accordingly, studies reported that Rab7 CMT2B variants exhibit an altered affinity towards GTP and GDP, leading to a higher rate of GDP/GTP exchange and increased GTP hydrolysis [47,48,49]. Our results shown in Figure 3A suggested that Rab7 GTPase activity is required for its RNF167-mediated ubiquitination; therefore, we investigated whether RNF167-mediated ubiquitination of Rab7 is affected in CMT2B variants. HEK293T cells were transfected with plasmids encoding GFP-tagged Rab7 WT or CMT2B variants in the presence or absence of exogenous RNF167. After lysis, an anti-GFP immunopurification of Rab7 proteins was performed from the total cell lysate before the IP complexes were resolved on SDS-PAGE and immunoblotted against UB. The results show that RNF167-mediated ubiquitination of Rab7 CTM2B variants was reduced when compared with RNF167-mediated ubiquitination of WT Rab7 (Figure 6B). This indicates that the tested CTM2B variants exhibit impaired ubiquitination. We have demonstrated that RNF167-dependant ubiquitination of Rab7 leads to vesicle enlargement (Figure 2F); therefore, we first determined the diameter of CTM2B-variant-positive vesicles co-transfected with a control vector in HeLa cells. As expected for mutants with putative reduced ubiquitination, vesicles positive for Rab7 CMT2B variants, but not Rab7 L129F, were significantly smaller than those of Rab7 WT (WT, 0.815 ± 0.008; K126R, 0.759 ± 0.008, *p* = 0.0026; L129F, 0.791 ± 0.010, *p* = 0.8835; K157N, 0.723 ± 0.007, *p* < 0.0001; N161I, 0.712 ± 0.010, *p* < 0.0001; V162M, 0.727 ± 0.008, *p* < 0.0001) (Figure 6C,D), indicating that expression of Rab7 CMT2B variants led to smaller vesicles. In contrast, the co-expression of RNF167 with either Rab7 WT or CTM2B variants in HeLa cells significantly enlarged the vesicles compared with conditions without RNF167 (WT + RNF167, 0.978 ± 0.011, *p* < 0.0001; K126R + RNF167, 0.859 ± 0.014, *p* < 0.0001; L129F + RNF167, 0.957 ± 0.013, *p* < 0.0001; K157N + RNF167, 0.893 ± 0.010, *p* < 0.0001; N161I + RNF167, 0.924 ± 0.013, *p* < 0.0001; V162M + RNF167, 0.896 ± 0.011, *p* < 0.0001) (Figure 6C,D). These data support the idea that RNF167 could restore the defect in vesicle size of many Rab7 CMT2B variants. Finally, the Manders’ overlap coefficient between Rab7 CMT2B variants and RNF167 showed that only the Rab7 K126R mutant had a higher colocalization with RNF167 than Rab7 WT or any other CMT2B variant (WT, 0.089 ± 0.010; K126R, 0.161 ± 0.014, *p* = 0.0002; L129F, 0.114 ± 0.012, *p* > 0.9999; K157N, 0.090 ± 0.010, *p* > 0.9999; N161I, 0.109 ± 0.012, *p* > 0.9999; V162M, 0.133 ± 0.018, *p* = 0.4185) (Figure 6E).

## 3. Discussion

Numerous mechanisms are employed to maintain a functional cellular proteome. Among the various dynamic systems maintaining proteostasis, post-translational ubiquitination regulates protein degradation, protein–protein interaction, DNA repair, protein endocytosis, and membrane trafficking [50,51,52]. Ubiquitination also controls late endosome and lysosome positioning, and the transmembrane ubiquitin E3 ligase RNF167 has been implicated in these processes [41,45]. Here, we show that Rab7 protein, a Rab GTPase involved in late endosome and lysosome positioning, is ubiquitinated by RNF167. We also demonstrate that GTPase activity and membrane anchoring are required for RNF167-mediated Rab7 ubiquitination.

This study reports mechanistic insights for the ubiquitination of Rab7 as a newly identified substrate for RNF167. It demonstrates that Rab7 arginine substitutions of lysines 191 and 194, which similarly affect Rab7 function in the cell [20,22], completely abrogated RNF167-mediated ubiquitination. These results corroborate evidence from another study that reported the major ubiquitination sites of Rab7 were K191 and K194 [22]. For instance, the Rab7 2KR variant has a reduced presence in membrane fractions following cell fractionation [21]. This corresponds with the action of the deubiquitinating enzyme USP32 on the UB-conjugated K191 of Rab7, which releases Rab7 from the endosomal membrane and reduces the size of vesicles containing Rab7 [21]. Although the opposing roles of RNF167 and USP32 likely control Rab7 function, our results show that the RNF167-mediated ubiquitination of Rab7 protein increases the association of UB-conjugated Rab7 with late endosomes, which, in turn, results in an endosomal vesicle enlargement. Our data implicate the exchange of GDP for GTP by the Rab7 protein as an important feature of the regulatory mechanism of Rab7 ubiquitination and membrane association. Accordingly, our results reveal that the ubiquitination of Rab7 protein by RNF167 is only possible after Rab7 guanosine nucleotide exchange, demonstrated by the lack of Rab7 ubiquitination of the inactive T22N Rab7 mutant. Therefore, the mechanism underlying RNF167-mediated Rab7 ubiquitination likely requires that cytosolic GDP-bound inactive form of the Rab7 protein must be prenylated to anchor at the endosomal membrane [53,54]. This allows the exchange of GDP for GTP by a Rab7-activating GEF and availability for ubiquitination by the transmembrane UB ligase RNF167. The Rab7-prenylation-deficient variant (C205S/C207S) is not ubiquitinated by RNF167 (Figure 2B); thus, our results strongly support the hypothesis that prenylation and the membrane anchoring of Rab7 are critical parts of the mechanism leading to its RNF167-mediated ubiquitination. Together, our results indicate that RNF167 controls Rab7 membrane association once the GTPase has been activated.

This study shows that RNF167 and Rab7 proteins affect each other’s subcellular localization to lysosomes, but do not regulate each other’s lysosome positioning functions. Rab7 is a central protein involved in the process of regulating late endosome to lysosome fusion [13,55]; therefore, our results which showed that knocking down Rab7 led to a decreased overlap of RNF167 and Lamp1 were expected. RNF167-mediated Rab7 ubiquitination does not occur on lysosomes but on late endosomes, as indicated by the increased overlap between Rab7 and Lamp1 in RNF167-depleted cells, suggesting that Rab7 mediates late endosome fusion with lysosomes when it is not UB-modified by RNF167. This is supported by a study indicating that the Rab7 2KR mutant shows an increased binding with its effector RILP [21]. RILP recruitment to the membrane by Rab7 leads to HOPS complex recruitment that induces late endosome and lysosome fusion [33].

We recently reported that the RNF167 RING-domain inactive variant H250W/H253W interacts better with its substrate GluA2, an AMPA-type glutamate receptor subunit, than WT RNF167 [42]. In the present study, we show that Rab7 binding is more efficient with DN RNF167 than with WT RNF167, indicating that substrate ubiquitination likely inhibits RNF167-Rab7 interaction. This could explain why we measured an increased overlap between Rab7 and DN RNF167 in HeLa cells. It could also be argued that the increased binding between DN RNF167 and Rab7 is the result of the inability of RNF167 to ubiquitinate Rab7, thus suggesting that the ubiquitination of Rab7 protein weakens the binding of RNF167 with Rab7. Although we do not yet possess the answer to such an intriguing possibility, the appropriate targeting of active Rab7 protein to the endosomal membrane is likely one of the most biologically important steps in the mechanism of its ubiquitination.

Based on the results presented, we uncovered a molecular mechanism requiring Rab7 membrane targeting and GTPase activity for RNF167-mediated Rab7 ubiquitination that regulates endosomal vesicle size. Importantly, this study demonstrates that the substitution of amino acid residues within the GTPase domain of Rab7 protein impairs its ubiquitination, exemplified by CMT2B Rab7 variants that exhibit an altered RNF167-mediated ubiquitination. CMT2B substitutions are mostly localized in a region of the protein involved in or close to amino acid residues participating in guanine-nucleotide-binding. This lets us propose that mutation within the GTPase domain of the *RAB7A* gene would encode a protein with an altered conformation, and thus partially inhibit the mechanisms leading to the ubiquitination of the Rab7 protein. One example supporting this idea is the altered conformation of the guanine-nucleotide-binding pocket in the L129F variant. This alteration leads to an increase in nucleotide-binding pocket size, explaining the observed higher dissociation rate of Rab7 with GDP versus GTP. In addition, the crystal structure of Rab7 indicates that K157, another residue identified as being substituted in CMT2B, directly binds to the guanine nucleotide, thus providing a plausible mechanistic explanation as to how the CMT2B Rab7 K157N variant leads to a higher dissociation rate [49]. Furthermore, other studies indicate that Rab7 CMT2B variants have higher dissociation kinetics for both GTP and GDP, and slower GTP hydrolysis. Accordingly, because there is a higher concentration of free GTP in CMT2B variant cells, researchers suggested that Rab7 mutants are overactivated because they have more chance to be associated with GTP [48,49,56]. In contrast, a study using *Drosophila* states that CMT2B amino acid substitutions (L129F, K157N, N161T and V162M) are only partially functional [57]. Specifically, the study showed a decrease in Rab7 protein recruitment to endosomes and that the presence of Rab7 variants in transgenic flies was enough to rescue Rab7 function in the cell [57]. Additionally, the newly identified Rab7 K126R variant reduces EGFR degradation, further supporting the partial function of this CMT2B variant [47].

Despite the discrepancies in the literature, studies support a rapid GDP and GTP dissociation in these CMT2B variants [47,48,49,56], which would explain the observed reduced Rab7 CMT2B variant ubiquitination by RNF167. Nonetheless, our results interestingly show that RNF167 UB-ligase activity increases Rab7′s ubiquitination; thus, higher RNF167 activity could potentially reverse Rab7 CMT2B variants’ cellular phenotypes, such as reduced vesicle size. Notably, previous evidence showed that Rab7-mediated endosomal membrane tubulation is involved in the recycling of membrane proteins through the TGN [58]. Now, we show that the activity of the endolysosomal UB-ligase RNF167 affects the endosomal membrane localization of Rab7 protein. Future studies should investigate RNF167′s function regarding the molecular impairments of Rab7 CMT2B variants in membrane tubulation and the consequence of endosomal protein recycling back to the plasma membrane in neurons.

## 4. Materials and Methods

### 4.1. Molecular Biology

pEGFP-C1-Rab5, pEGFP-C1-Rab7, pEGFP-C1-Rab9 and pEGFP-C1-Rab11, pcDNA3.1D RNF167-V5-His WT and H250W/H253W were used previously [40]. cDNA encoding 2×HA-hRab7A WT was sequence-optimized, synthesized then cloned into pUCIDT-KAN at IDT (Integrated DNA Technologies, Coralville, IA, USA). pUCIDT-KAN 2×HA-hRab7A plasmid was digested using EcoRI and XhoI before ligating the resulting 2×HA-hRab7A fragment into pcDNA3.1 (Invitrogen, ThermoFisher Scientific, Burlington, ON, Canada). The pUCIDT-KAN 2×HA-hRab7A plasmid, that contains a BamHI site between the 2×HA and hRab7A cDNA, was cut with BamHI and SalI and ligated into BglII and SalI sites in pEGFP-C1 (Takara Bio USA, Inc., San Jose, CA, USA) or into BamHI and SalI of pGEX-4T-1 (Cytiva, Mississauga, ON, Canada). pET-52b(+) HA-RNF167-6xHis (a.a. 195–350), either WT or H250W/H253W, were obtained from GenScript (Piscataway, NJ, USA).

Mutagenesis of cDNA encoding hRab7A cloned into pUCITD-KAN, pcDNA3.1 or pEGFP-C1 was performed with the following primers synthetized by IDT: K38R (forward: 5′-caagaaattcagcaaccagtatagggccaccattgg-3′, reverse: 5′-ccaatggtggccctatactggttgctgaatttcttg-3′), 2KR (forward: 5′-gtttccagagcctatcaggctggataggaatgatagagccaagg-3′, reverse: 5′-ccttggctctatcattcctatccagcctgataggctctggaaac-3′), T22N (forward: 5′-gacagcggcgtgggcaagaactccctgatgaac-3′, reverse: 5′-gttcatcagggacttcttgcccacgccgctgtc-3′), Q67L (forward: 5′-gacaccgccggcctggagagattccaat-3′, reverse: 5′-attggaatctctccaggccggcggtgtc-3′), C205S/C207S (forward: 5′-gcgccgagagcagtagcagctgagtcgac-3′, reverse: 5′-gtcgactcagctgctactgctctcggcgc-3′), K126R (forward: 5′-ccttcgtggtgctgggaaatagaatcgacctgga-3′, reverse: 5′-tccaggtcgattctatttcccagcaccacgaagg-3′), L129F (forward: 5′-tgctgggaaataaaatcgacttcgaaaatagacaggtggctac-3′, reverse: 5′-gtagccacctgtctattttcgaagtcgattttatttcccagca-3′), K157N (forward: 5′-acttcgagacaagcgctaatgaggccatcaacg-3′, reverse: 5′-cgttgatggcctcattagcgcttgtctcgaagt-3′), N161I (forward: 5′-caagcgctaaagaggccatcatcgtggaacagg-3′, reverse: 5′-cctgttccacgatgatggcctctttagcgcttg-3′) and V162M (forward: 5′-ctaaagaggccatcaacatggaacaggcttttcag-3′, reverse: 5′-ctgaaaagcctgttccatgttgatggcctctttag-3′). The integrity of all plasmids encoding Rab7 was confirmed by Sanger sequencing (Genome Quebec, Montreal, QC, Canada).

### 4.2. Silencing of RAB7A and RNF167 Genes

The TriFECTa Dicer Substrate duplex RNAi kit against human *RAB7A* (#1: 5′-rArArCrCrArGrUrArUrGrUrGrArArUrArArGrArArArUrUCA-3′ and 5′-rUrGrArArUrUrUrCrUrUrArUrUrCrArCrArUrArCrUrGrGrUrUrCrA-3′, #2: 5′-rGrUrGrCrUrArCrArGrCrArArArArArCrArArCrArUrUrCCC-3′ and 5′-rGrGrGrArArUrGrUrUrGrUrUrUrUrUrGrCrUrGrUrArGrCrArCrCrA-3′, #3: 5′-rCrCrGrUrUrArGrArUrCrArGrCrArUrUrCrUrArCrUrArCAA-3′ and 5′-rUrUrGrUrArGrUrArGrArArUrGrCrUrGrArUrCrUrArArCrGrGrGrA-3′) and human RNF167 (#1: 5′-rArArCrUrUrUrGrArCrCrUrCrArArGrGrUrC-3′ and 5′-rCrArUrUrUrArGrGrArCrCrUrUrGrArGrGrU-3′, #2: 5′-rUrCrGrArCrUrUrArCrCrArArArGrArGrCrA-3′ and 5′-rUrUrUrCrArGrUrUrGrCrUrCrUrUrUrGrGrU-3′, #3: 5′-rGrUrCrUrUrCrArCrUrUrCrUrUrGrGrGrCrU-3′ and 5′-rUrUrUrUrArUrUrArGrCrCrCrArArGrArArG-3′) was purchased from IDT. The negative control scramble DsiRNA (cat. #DSNC1) was used.

### 4.3. Antibodies

Antibodies used for Western blot (WB) and immunofluorescence (IF) were as follows: mouse anti-ubiquitin (clone P4D1, 1:750, Santa Cruz Biotechnology, Dallas, TX, USA, cat. #sc-8017, RRID: AB_2762364), mouse anti-GFP (1:1000, Cell Signaling Technologies, New England Biolabs, Whitby, ON, Canada, cat. #2955S, RRID: AB_1196614), mouse anti-V5 (1:1000, Initrogen, ThermoFisher Scientific, cat. #46-0705), rabbit anti-HA (1:1000, Cell Signaling Technologies, New England Biolabs, cat. #3724, RRID: AB_1549585), rabbit anti-GST (1:1000, Cell Signaling Technologies, New England Biolabs, cat. #2625, RRID: AB_490796) mouse anti-HA (clone HA.11, 1:1000; BioLegend, San Diego, CA, USA, cat. #901502, RRID: AB_2565007), mouse anti-GAPDH (clone 8C2, 1:500, Abnova, Walnut, CA, USA, cat. #MAB0687, RRID: AB_1204387), mouse anti-Transferrin receptor (1:500, ThermoFisher Scientific, cat. #13-6800, RRID: AB_2533029), rabbit anti-Rab7 (1:1000, Cell Signaling Technologies, New England Biolabs, cat. #9367, RRID: AB_1904103), rabbit anti-Lamp1 (IF: 1:500, WB: 1:1000; Cell Signaling Technologies, cat. #9091, RRID: AB_2687579), horseradish peroxidase (HRP)-conjugated horse anti-mouse (1:10,000; Cell Signaling Technology, New England Biolabs, cat. #7076, RRID: AB_330924), HRP-conjugated goat anti-rabbit (1:10,000; Cell Signaling Technologies, New England Biolabs, cat. #7074, RRID: AB_2099233), Alexa Fluor 488-conjugated goat anti-rabbit (1:1000, Thermo Fisher Scientific, Waltham, MA, USA, cat. #A11070, RRID: AB_142134), and Alexa Fluor 647-conjugated goat anti-rabbit (1:1000, Thermo Fisher Scientific, cat. #A31626).

### 4.4. Cell Culture and Transfection

Human embryonic kidney HEK293T/17 cells were purchased from the American Type Culture Collection (ATCC) (Gaithersburg, MD, USA, cat. #CRL-11268, RRID: CVCL_1926) and human cervical adenocarcinoma HeLa cells were kindly provided by Diana Alison Averill lab (UQAM, Département des Sciences Biologiques, Montréal, QC, Canada). Cells were maintained in Dulbecco’s modified Eagle’s medium (DMSO) (ThermoFisher Scientific, cat. #11995-065) with 10% fetal bovine serum (FBS) (VWR Life Science, Radnor, PA, USA, cat. #CA45001-106) at 37 °C and 5% CO_2_. HEK293T/17 cells were transfected with Lipofectamine 2000 (ThermoFisher Scientific, cat. #11668019) reagent as described previously [59]. Briefly, Lipofectamine 2000 was diluted in Opti-MEM, incubated for 5 min before mixing with plasmid DNA and incubated for 25 min, using a ratio of 2 µL of Lipofectamine 2000 for 1 µg of total plasmid DNA for each 1 × 10^6^ plated cells. HEK293T/17 cells were added to the mixture of DNA–Lipofectamine 2000 in a poly-D-lysine-treated plate before incubating for 24 h at 37 °C with 5% CO_2_. 24 h before transfection, 3 × 10^4^ HeLa cells were plated in the 24-wells plate. Cells were transfected using standard calcium phosphate protocol. Briefly, 0.4 µg total plasmid DNA was incubated for 3 min with 250 mM CaCl_2_ before mixing one droplet at a time with an equal volume of 2× Hanks’ Balanced Salt Solution (HBSS) to form a calcium phosphate–DNA precipitate. After 30 min incubation, the mixture was added to the cells and incubated for 24 h at 37 °C with 5% CO_2_. For gene silencing, HeLa cells were plated 4 h before transfection at either 1.25 × 10^4^ cells/well of a 24-well plate (for microscopy) or 2.5 × 10^5^ cells/well of a 6-well plate (for Western blot analysis). Then, 0.5 µL/well for a 24-well plate or 2 µL/well for a 6-well plate of Lipofectamine 2000 were diluted and incubated for 5 min in Opti-MEM medium before mixing with DsiRNA. The mixture was incubated for 20 min before adding to the cells at a final concentration of 10 nM DsiRNA in each well. Cells were transfected using standard calcium phosphate protocol (24 h transfection) 48 h after DsiRNA transfection, as described above.

### 4.5. Immunoprecipitation and Co-Immunoprecipitation

For immunoprecipitation (IP), transfected HEK293T/17 cells were washed with ice-cold PBS (10 mM Na_2_HPO_4_, 2 mM KH_2_PO_4_, 137 mM NaCl, 2.7 mM KCl, pH 7.4) then lysed in ice-cold RIPA buffer (20 mM Tris–HCl, 150 mM NaCl, 1% Triton X-100, 0.1% Sodium Dodecyl Sulfate (SDS), 0.5% Sodium Deoxycholate, 5 mM Ethylenediaminetetraacetic acid (EDTA), 20 mM N-ethylmaleimide (NEM, MilliporeSigma, Oakville, ON, Canada), 50 µM PR-619 (MilliporeSigma, cat. #662141), 1× protease inhibitor cocktail without EDTA (Bimake.com, Houston, TX, USA, cat. #B14001), pH 7.5). The total lysate was obtained by centrifuging at 21,000× *g* for 15 min at 4 °C, then slowly rotated overnight at 4 °C with Protein A/G Plus Agarose beads (Santa Cruz Technology, Dallas, TX, USA, cat. #sc-2003) and 2 µL of anti-GFP Rabbit Serum (ThermoFisher Scientific, cat. #A6455, RRID: AB_221570). For co-immunoprecipitation (Co-IP), cells were lysed in Co-IP buffer (20 mM Tris–HCl, 150 mM NaCl, 1% Triton X-100, 1× protease inhibitor cocktail without EDTA (Bimake.com), pH 7.5) and slowly rotated for 2 h at 4 °C with 10 µL of mouse anti-HA affinity gel (Biotool.com, Jupiter, FL, USA, cat. #B23302). Immunoprecipitated proteins were washed with lysis buffer and eluted with Laemmli 2× buffer (100 mM Tris–HCl, pH 6.8, 20% glycerol, 2% SDS, 0.02% p/v bromophenol blue, 10% β-mercaptoethanol) for 5 min at 95 °C.

### 4.6. Protein Expression and Purification

Proteins were purified as described previously [42]. Briefly, plasmids encoding HA-RNF167-6xHis WT or H250W/H253W variant (a.a. 194–350) were transformed in *E. coli* BL21 DE3 pLysS bacterial cells, grown at 28 °C for 16 h in LB media containing 100 µg/mL Ampicillin and 50 µg/mL chloramphenicol (LB-AMP-CHL). The cells were used to inoculate a new flask containing LB-AMP-CHL for growth at 37 °C until the optical density at 600 nm reached 0.6. Protein production was induced with the addition of 1 mM Isopropyl β-D-1-thiogalactopyranoside (IPTG) for 2 h at 30 °C. Then, 1 µM ZnSO_4_ was added during protein synthesis to maximize RNF167′s RING domain structure and stability. For GST-Rab7 protein production, a similar protocol was used, where transformed bacterial cells were grown in LB-AMP media containing 2% glucose without the addition of ZnSO_4_, and protein production used 0.5 mM IPTG for induction. After protein synthesis, cells were pelleted by centrifugation (3000× *g*, 20 min, 4 °C), and either lysed immediately for purification, or flash-frozen in liquid nitrogen before storage at −80 °C until further processing.

For HA-RNF167-6xHis WT purification, pelleted cells were suspended in ice-cold His-tag purification buffer (20 mM phosphate buffer, 500 mM NaCl, 0.5% Triton X-100, 20 mM imidazole, 0.5 mM 1,4-Dithiothreitol (DTT), 1 mM phenylmethylsulphonyl fluoride (PMSF), pH 7.4) and slowly rotated (15 rpm, 20 min, 4 °C) for lysis before clearing the total lysate by centrifugation (16,000× *g*, 30 min, 4 °C). Filtered supernatants containing soluble proteins were loaded on HisTrap FF crude column preequilibrated with His-tag washing buffer (20 mM phosphate, 500 mM NaCl, 20 mM imidazole, pH 7.4) using the AKTA Start Fast Protein Liquid Chromatography (FPLC) system (Cytiva). After protein injection and purification, the column was washed sequentially with buffers containing 20 and then 60 mM imidazole before protein elution with buffer (20 mM phosphate, 500 mM NaCl, containing 500 mM imidazole, pH 7.4). HA-RNF167-6xHis RING mutant H250W/H253W producing cells were lysed (20 mM Tris–HCl, 350 mM NaCl, 1% Triton X-100, 20 mM imidazole, 1 mM PMSF, pH 8.0) to collect inclusion bodies by centrifugation (16,000× *g*, 30 min, 4 °C). Pelleted inclusion bodies suspended in lysis buffer (20 mM Tris–HCl, pH 8.0, 350 mM NaCl, 6 M Urea, 20 mM imidazole, 1 mM PMSF) were slowly rotated (15 rpm, 1 h, 4 °C) before sonication (5 × 10 s) and centrifugation (16,000× *g*, 30 min, 4 °C). Extracted proteins in the supernatant were added to Ni-NTA beads (Qiagen, Toronto, ON, Canada, cat. #30210) pre-equilibrated in washing buffer (20 mM Tris–HCl, 350 mM NaCl, 20 mM imidazole) before being slowly rotated (15 rpm, overnight, 4 °C). Protein purification was processed using gravity flow and RNF167 mutant protein was refolded on beads using 15 column volumes (CV) of His-tag washing buffers supplemented with urea using a stepwise removal (urea: 4 M, 2 M, and 0 M). Refolded purified proteins were eluted using elution buffer containing 500 mM imidazole, and fractionated proteins were concentrated using Pierce protein concentrator PES (10K MWCO, ThermoFisher Scientific). Purified proteins were dialyzed overnight in PBS containing 1 µM ZnSO_4_, quantified using the Pierce BCA Protein Assay kit (Thermo Fisher Scientific) before flash-freezing in liquid nitrogen and stored at −80 °C until use for the in vitro assay.

For GST-Rab7 protein purification, cells were suspended in GST lysis buffer (20 mM Tris–HCl, 150 mM NaCl, 1% Triton X-100, 5 mM EDTA, 1 mM DTT, 1 mM PMSF, pH 8.0) and the soluble total lysate was obtained as described for WT RNF167. The total lysate was loaded on a GSTrap 4B column equilibrated with GST washing buffer (20 mM Tris–HCl, 150 mM NaCl, pH 8.0). After lysate injection into the FPLC system, the column was washed with GST washing buffer and eluted with GST elution buffer (50 mM Tris–HCl, 20 mM reduced glutathione, pH 8.0). Purified proteins were dialyzed in GST dialysis buffer (20 mM Tris–HCl, 150 mM NaCl, 10% glycerol, 2 mM DTT, pH 8.0), quantified, and stored as described above.

### 4.7. In Vitro Ubiquitination Assay

The ubiquitination assay was performed as described previously [42]. Briefly, 100 nM UBE1 (Boston Biochem, Cambridge, MA, USA, cat. #E-304), 50 µM UB, 2 mM ATP, 2 µM 6xHis-UBE2D1 WT (Boston Biochem, cat. #E2-616) or C85A variant (Enzo life sciences, Farmingdale, NY, USA, cat. #BML-UW 9055), 4 µM purified HA-RNF167-6xHis WT or H250W/H253W variant, and 2 µM purified GST-Rab7 were mixed in 1× reaction buffer (Boston Biochem), and incubated for 1 h at 30 °C. Half of the reaction was quenched with Laemmli 2× buffer and the other half was diluted in GST pull-down buffer (20 mM Tris–HCl, 150 mM NaCl, 1% Triton X-100, pH 8.0) for GST-Rab7 protein enrichment using Glutathione Sepharose 4B resin (Cytiva), pre-equilibrated in GST pull-down buffer containing 1% BSA to block non-specific interaction, by slow rotation for 1 h at 4 °C. The resin was washed with GST pull-down buffer and proteins were eluted with Laemmli 2× buffer, followed by 5 min incubation at 95 °C. Subsequently, 25% reaction was loaded on SDS-PAGE for anti-GST Western blotting and 100% of purified GST-Rab7 was loaded for anti-P4D1 Western blotting.

### 4.8. Subcellular Fractionation

The separation of membranes and cytosol was performed as described previously [21]. Briefly, 6 × 10^6^ HEK293T/17 cells were transfected 24 h prior to fractionation. After PBS washes, cells were recovered in ice-cold fractionation buffer 1 (20 mM Tris–HCl, 1 mM MgCl_2_, 1× protease inhibitor, pH 7.5) and the cellular suspension was passed through a G21 needle at least 20 times and until the solution was clear. Whole cells and debris were pelleted by centrifugation (1000× *g*, 10 min, 4 °C) and the supernatant was centrifuged once more (2000× *g*, 20 min, 4 °C). The cytosolic fraction was separated from the membrane fraction by centrifugation (20,000× *g*, 30 min, 4 °C) (supernatant = cytosolic, pellet = membranes). The pelleted membranes were finally washed with ice-cold fractionation lysis buffer 1 and the centrifugation was repeated. The pellet was solubilized in fractionation buffer 2 (20 mM Tris–HCl, 0.5% NP-40, 1 mM MgCl_2_, 1× protease inhibitor, pH 7.5). Total protein in each fraction was quantified using the Pierce BCA Protein Assay kit. A 5 µg sample of total protein was loaded on the gel for separation before Western blotting for transferrin receptor, tubulin and GAPDH, whereas 1.25 µg of total protein was used for transfected protein detection.

### 4.9. Western Blot

Proteins were separated on SDS-PAGE supplemented with 0.5% of 2,2,2-Trichloroethanol to allow the stain-free visualization of proteins migrated at 40 mA/gel. Using the stain-free option in the Image Lab software (RRID: SCR_014210) of the ChemiDoc MP Imaging System (Bio-Rad, Hercules, CA, USA, RRID: SCR_019037) enabled the visualization of proteins in gel or on the 0.45 µm PVDF membrane after protein transfer using Trans-blot Turbo (Bio-Rad, Mississauga, ON, Canada). The membrane was blocked with 5% skim milk dissolved in TBS-T buffer (20 mM Tris–HCl, 140 mM NaCl, 0.3% Tween-20) before incubation with primary antibody diluted in TBS-T with 0.5% NaN_3_ for more than 1 h, followed by 1 h incubation with HRP-conjugated secondary antibody diluted in TBS-T. Immunoblotted proteins were visualized using ECL substrate (Bio-Rad) with ChemiDoc MP Imaging System (Bio-Rad) using *chemi* function in Image Lab software version 6.0.1 (Bio-Rad).

### 4.10. Western Blot Quantification

Band intensity quantification was performed with Image Lab software version 6.0.1 (Bio-Rad). Western blot signal intensity was normalized using stain-free signal intensity on the PVDF membrane. To do so, the lane and bands function in the software was used to delimit the Western blot band and the stain-free bands. For stain-free signal quantification, the same number of bands was quantified for each lane in the same conditions. The band intensity was compiled in Excel for normalization calculations. Statistical analyses and graphics were made in GraphPad Prism version 9 (San Diego, CA, USA, RRID: SCR_002798).

### 4.11. Immunofluorescence Microscopy

HeLa cells were plated on round 12 mm glass coverslip #1.5 (Electron Microscopy Science, Hadfield, PA, USA, cat. #71887-04) in a 24-well plate for 24 h and transfected using calcium phosphate protocol, as described above, 24 h before immunofluorescence. All manipulations were performed under dimmed light and at room temperature. Coverslips were washed with PBS and then fixed with PFA solution (4% PFA, 4% sucrose in PBS) for 15 min. After washing away the fixative solution, membrane permeabilization was performed with 0.25% Triton X-100 in PBS for 15 min. Coverslips were blocked for 1 h with 10% normal goat serum (NGS) diluted in PBS before 1 h incubation with primary antibody diluted in 3% NGS in PBS. Coverslips were washed and then incubated for 1 h with secondary antibody diluted in 3% NGS in PBS. Coverslips were washed and then stained with a solution of 2 µg/mL DAPI in PBS for 5 min before mounting on a slide with ProLong Diamond Antifade (Life Technologies, Rockville, MD, USA, cat. #P36965). The coverslips were cured overnight before image acquisition. Three non-blinded experiments were performed for all assays.

The images were acquired using an inverted epi-fluorescence microscope: Olympus IX83 (RRID: SCR_020344) equipped with a U Plan S-Apo 60×/1.35 numerical aperture oil objective, an X-Cite Xylis 365 LED-based illumination source (Excelitas Technologies Corp., Waltham, MA, USA) and a Zyla 4.2 Plus sCMOS camera (Andor, Oxford, UK). Olympus CellSens Dimension software version 2.2 (Olympus, Toronto, ON, Canada) (RRID: SCR_014551) was used for image acquisition. Images, acquired as z-stack with 0.27 µm intervals, were deconvoluted using the Olympus 3D Deconvolution function in the Olympus CellSens Dimension software version 2.2 (Olympus).

For vesicle diameter measurement, one single plane was chosen for each image and the five largest vesicles positive for Rab7 (with RNF167 when co-expressed) were measured with the Arbitrary line function in the Olympus CellSens Dimension software version 2.2 (Olympus).

For Manders’ overlap coefficient measurement, one single plane from the z-stack was chosen for each image and, using the JACoP plugin in Fiji, the threshold was adjusted to obtain single puncta signal for every single cell using the function in the plugin before measuring the overlap coefficient.

For quantification of the Lamp1 signal distribution in the cell, analysis was performed on one single Z-plane. A line was traced from the end of the nucleus (DAPI staining) to the plasma membrane (the end of the cell) using the straight-line function in Fiji. The keyboard function CTRL + K allowed obtaining the fluorescence intensity graphic in relation to the distance, where 0 µm was the end of the nucleus and the highest distance was the end of the cell. The ratio of the perinuclear/peripheral Lamp1 fluorescence signal was calculated with Python coding, and the code ran on Anaconda JupiterLab Version 3.2.1. The code separated the total distance of the line in ten equal segments (1), where the first segment was the closest to the nucleus and the tenth segment was the closest to the plasma membrane. The code then calculated the average of Lamp1 fluorescence intensity on the line for the second and ninth segment (2), which represented the perinuclear and the peripheral signals, respectively. Finally, the code divided the average intensity for the second segment with the average intensity for the ninth segment (3) to obtain the perinuclear/peripheral Lamp1 signal ratio. The detail of the code is provided in the Appendix A.

All data were compiled in Excel and statistical tests were performed using GraphPad Prism version 9.

### 4.12. Statistical Analysis

GraphPad Prism software version 9 was used for the following statistical analyses. Non-parametric one-way ANOVA with Kruskal–Wallis post-test and non-parametric *t*-test with Mann–Whitney post-test were used for the experiments with only one variable due to the non-Gaussian data distribution. Two-way ANOVA with Tukey’s post hoc test was used for experiments with two variables.

## Figures and Tables

**Figure 1 ijms-23-07847-f001:**
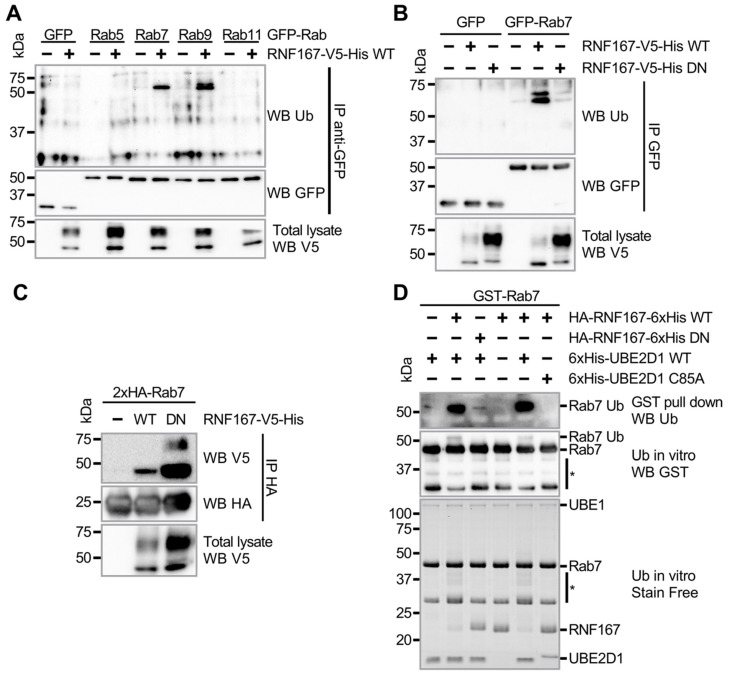
**Rab7 protein is a substrate of RNF167**. (**A**) Late endosome-associated Rab GTPases are substrates of RNF167. HEK293T-transfected cells were lysed before immunoprecipitation with a GFP antibody. Ubiquitination of IP complexes was detected using a UB antibody. Blots are representative results of two independent experiments. (**B**) Rab7 ubiquitination in relation to RNF167 activity. HEK293T cells were transfected with respective plasmids encoding indicated proteins. Cells were lysed and immunoprecipitation was performed with a GFP antibody. Blots are representative results of two independent experiments. (**C**) When exogenously co-expressed in HEK293T cells, Rab7 co-immunoprecipitated with RNF167, either wild-type (WT) or RING variant H250W/H253W (DN). Transfected HEK293T cells were lysed in non-denaturing conditions before performing anti-HA immunoprecipitation. The presence of RNF167 in the Rab7-IP complexes was detected with a V5 antibody. Blots are representative of two independent experiments. (**D**) In vitro ubiquitination assays were performed using purified GST-Rab7, UBE1, ATP, UB, purified WT or DN HA-RNF167-6xHis, and either UBE2D1 WT or a C85A inactive variant. After incubation, the reaction was incubated with GSH resin to enrich ubiquitinated Rab7. Reactions separated on SDS-PAGE were transferred onto a PVDF membrane before immunodetection, as indicated. Representative immunoblots are shown. The asterisk (*) represents a truncated GST-Rab7 protein.

**Figure 2 ijms-23-07847-f002:**
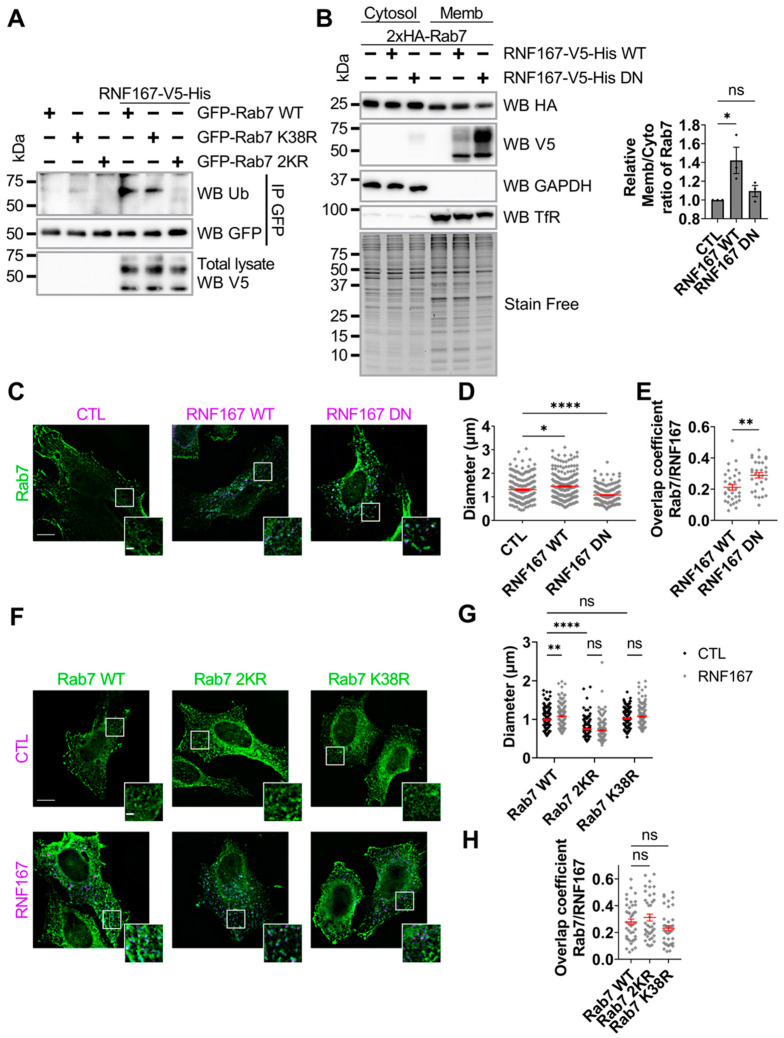
**RNF167 promotes Rab7 membrane localization**. (**A**) Lysine substitutions in Rab7 protein reduce RNF167-mediated ubiquitination. HEK293T cells were transfected with the respective plasmids encoding the indicated proteins indicated in panel (**A**). Cells were lysed before immunopurification with a GFP antibody. Rab7 variant 2KR represents a combination of both K191R and K194R. Blots are representative results of two independent experiments. (**B**) Subcellular fractionation to separate cell membrane from the cytosol from HEK293T cells transfected with plasmids encoding the indicated proteins. Fractions were immunoblotted with antibodies against HA (Rab7) and V5 (RNF167), GAPDH (cytosol marker), and Transferrin receptor (TfR, membrane marker). The stain-free signal was used as a loading control and used for normalization. The blots shown are representative results from three separate experiments. Non-parametric one-way ANOVA with Kruskal–Wallis post-test was used, ns = non-significant, *, *p* = 0.0459. (**C**–**E**) Quantitative analysis of vesicle diameter and overlap in HeLa cells expressing Rab7 and RNF167. HeLa cells were transfected to produce the indicated proteins, fixed, and labeled for immunofluorescence against HA-tag (Rab7, green) and V5-tag (RNF167, purple). (**D**) The sizes of the five largest vesicles corresponding to Rab7 and RNF167 signals were measured for each cell from three independent experiments. Non-parametric one-way ANOVA with Kruskal–Wallis post-test was used, ns = non-significant, *, *p* = 0.0177, ****, *p* < 0.0001. (**E**) The Manders’ overlap coefficient shows that RNF167 activity regulates Rab7 overlap. Non-parametric *t*-test with a Mann–Whitney post-test was used, ns = non-significant, **, *p* = 0.0025. (**F**–**H**) Quantitative analysis of vesicle diameter and overlap in HeLa cells expressing RNF167 with either WT, K38R or 2KR Rab7 constructs. (**G**) Five largest vesicles in HeLa cells exogenously producing both Rab7 and RNF167 proteins were measured in each cell from three independent experiments. Two-way ANOVA with Tukey’s post hoc test was used, ns = non-significant, **, *p* = 0.0028, ****, *p* < 0.0001. (**H**) The Manders’ overlap coefficient between RNF167 and Rab7. Non-parametric one-way ANOVA with Kruskal–Wallis post-test was used, ns = non-significant. For vesicle diameter: *n* = 38–44 cells corresponding to 190–220 vesicles analyzed. For Manders’ overlap coefficient: *n* = 30–42 cells. Scale bar = 10 µm and 2 µm for zoom image. Errors bars (red) represent mean ± SEM. All images are representative results of three independent experiments.

**Figure 3 ijms-23-07847-f003:**
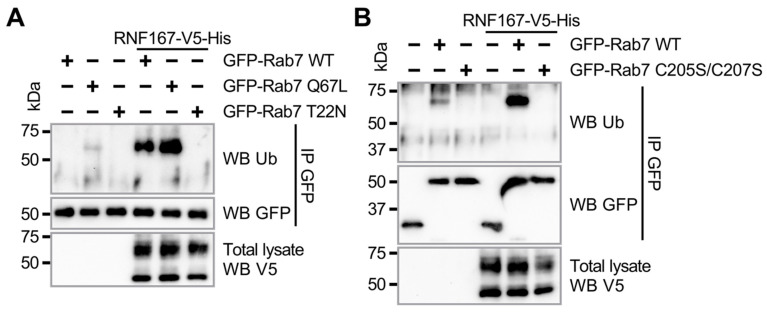
**RNF167-mediated ubiquitination of Rab7 requires GTPase activity and association with the membrane**. (**A**,**B**) HEK293T cells were transfected with plasmids encoding the indicated proteins, and cells were lysed before immunoprecipitation with a GFP antibody. Immunoblot with UB antibody following immunoprecipitation evaluated Rab7 ubiquitination. (**A**) Rab7 ubiquitination requires GTPase activity. (**B**) Rab7 ubiquitination requires its membrane anchoring. Blots are representative results of two independent experiments.

**Figure 4 ijms-23-07847-f004:**
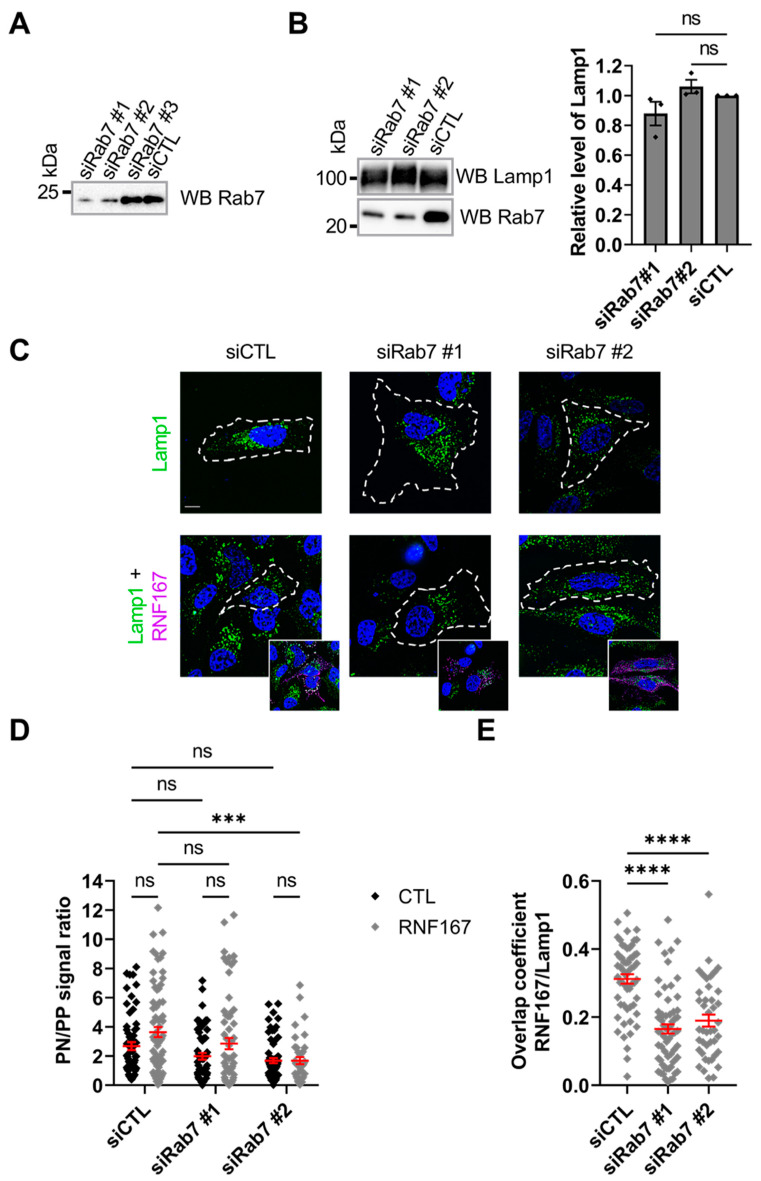
**Rab7 is involved in lysosome positioning in cells**. (**A**) Rab7 knockdown efficiency in HeLa cells transfected with DsiRab7s. Lysates were probed against endogenous Rab7 protein. siCTL = control DsiRNA. Blot is a representative result of three separate experiments. (**B**) Rab7 depletion does not alter endogenous Lamp1 protein abundance. DsiRab7-transfected HeLa cell lysates were probed for Lamp1 and Rab7 (**left**). Blots are representative results of three separate experiments. The stain-free signal (not shown) was used to normalize protein for quantitative analysis of Lamp1 abundance (**right**). siCTL = control DsiRNA. Non-parametric one-way ANOVA with Kruskal–Wallis post-test was used, ns = non-significant. (**C**–**E**) Impact of Rab7 level on RNF167 lysosomal positioning. (**C**) HeLa cells transfected with control or specific DsiRab7 and a control vector or WT RNF167-V5-His. Immunofluorescence assay was performed on fixed cells stained against V5-tag of RNF167 (purple), Lamp1 endogenous lysosome marker (green), and DAPI (blue) for the nucleus. Scale bar = 10 µm. Images are representative results of three separate experiments. (**D**) Quantification of lysosome positioning (perinuclear versus peripheral ratio) from the dataset presented in (**C**). PN = perinuclear, PP = peripheral. The jitter plot represents data from three independent experiments with *n* = 41–74 cells. Errors bars (red) represent mean ± SEM. Two-way ANOVA with Tukey’s post hoc test was used, ns = non-significant, ***, *p* = 0.0003. (**E**) Manders’ overlap coefficient analysis between RNF167 and Lamp1 in relation to the abundance of Rab7. The jitter plot represents data from 42 to 65 cells collected over three independent experiments. Errors bars (red) represent mean ± SEM. Nonparametric one-way ANOVA with a Kruskal–Wallis post-test was used. ****, *p* < 0.0001.

**Figure 5 ijms-23-07847-f005:**
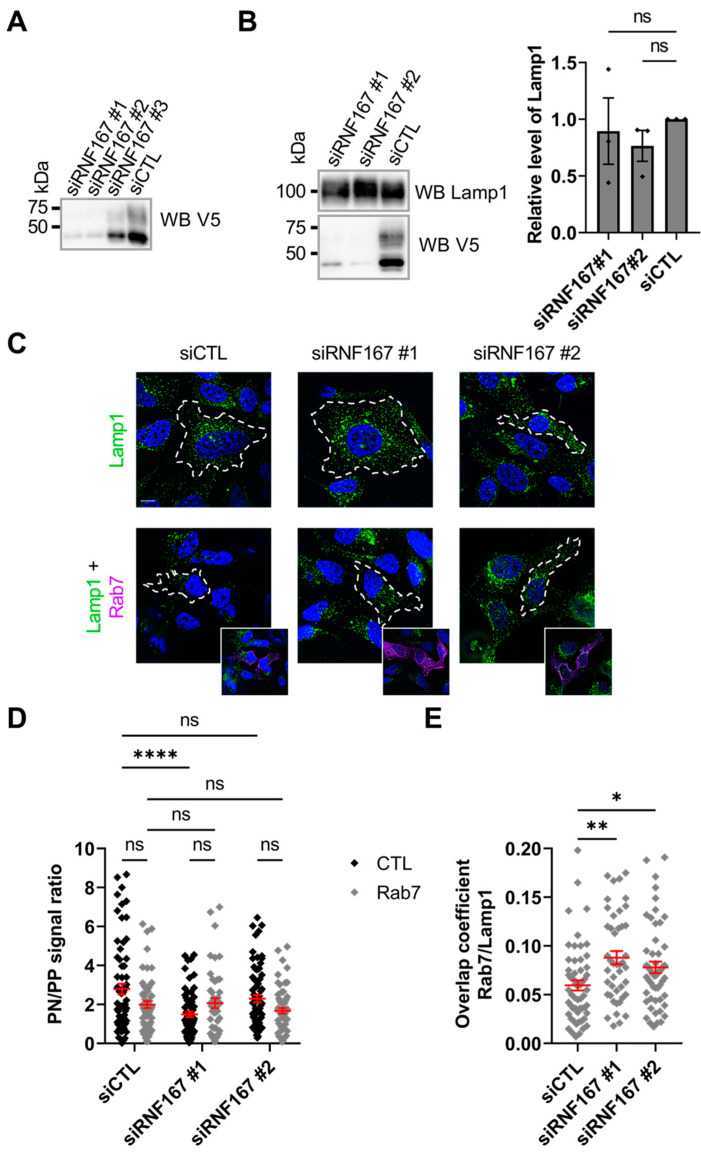
**Rab7 lysosome positioning is independent of RNF167 abundance**. (**A**) RNF167 DsiRNA knockdown efficiency was evaluated by the co-transfection of selected DsiRNF167 with RNF167-V5-His in HeLa cells. Cell lysates were probed with a V5 antibody against exogenous RNF167-V5-His. Blot is representative of results from three independent experiments. (**B**) RNF167 depletion does not alter endogenous Lamp1 protein abundance (**left**). Blots are representative results of three independent experiments. The stain-free signal (not shown) was used to normalize protein for quantitative analysis of Lamp1 abundance (**right**). siCTL = control DsiRNA. Non-parametric one-way ANOVA with Kruskal–Wallis post-test was used, ns = non-significant. (**C**–**E**) Impact of RNF167 expression on Rab7 lysosomal positioning. (**C**) Co-transfection of DsiRNF167 or its control in the presence or absence of 2xHA-Rab7. HeLa cells were fixed and labelled for the HA-tag of Rab7 (purple), endogenous Lamp1 lysosome marker (green) and DAPI (blue) for the nucleus. Scale bar = 10 µm. Images are representative results of three independent experiments. (**D**) Quantitation of perinuclear-to-peripheral Lamp1 signal ratio as explained in material and methods. PN = perinuclear, PP = peripheral. Data from three independent experiments totaling *n* = 46 to 71 cells. Errors bars (red) represent mean ± SEM. Two-way ANOVA with Tukey’s post hoc test was used, ns = non-significant, ****, *p* ˂ 0.0001. (**E**) Manders’ overlap coefficient between Rab7 and Lamp1. Errors bars (red) represent mean ± SEM. Non-parametric one-way ANOVA with Kruskal–Wallis post-test was used. *n* = 44 to 56 cells over three independent experiments, *, *p* = 0.0334, **, *p* = 0.0018.

**Figure 6 ijms-23-07847-f006:**
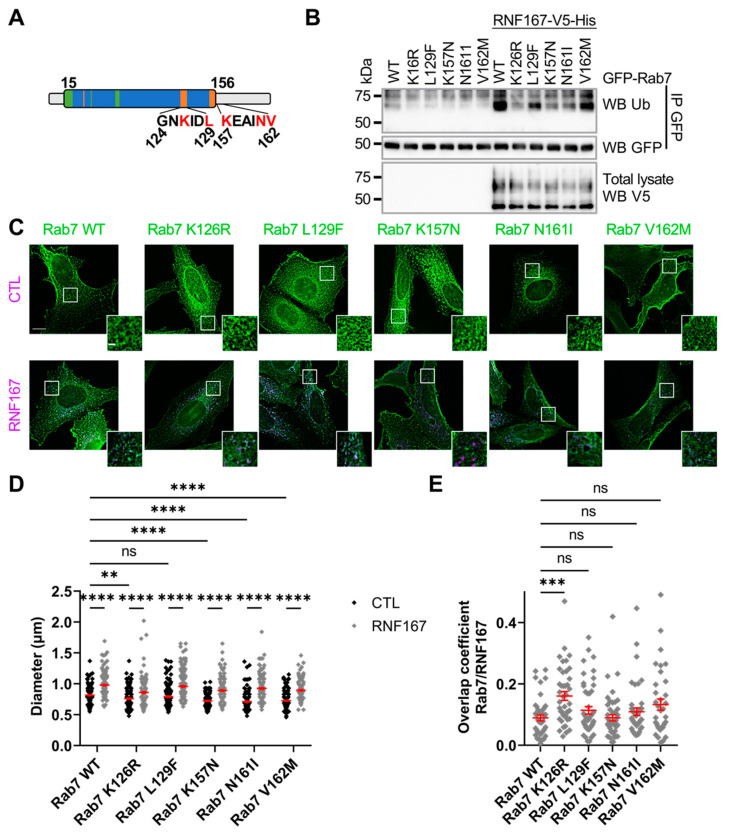
**Charcot–Marie–Tooth Type 2B variants of Rab7 impair RNF167-mediated ubiquitination**. (**A**) Schematic representation of Rab7 polypeptide linear structure and amino acid residues substituted in CMT2B variants. Blue, GTPase domain; green, magnesium binding region; orange, guanine binding region. Letters in red are amino acid residues substituted in CMT2B. (**B**) Evaluation of Rab7 CMT2B variant ubiquitination. HEK293T cells were transfected with plasmids encoding the indicated proteins and lysed for immunoprecipitation with GFP antibody. Immunoblot against UB evaluated Rab7 ubiquitination. Blots are representative results of two separate experiments. (**C**) HeLa cells were transfected with the indicated constructs and fixed for immunofluorescence against HA-tag (Rab7, green) and V5-tag (RNF167, purple). Scale bar = 10 µm and 2 µm for zoom image. Images are representative results of three separate experiments. (**D**) Measurement of vesicle diameter of Rab7 CMT2B variants. The diameters of the five largest vesicles positive for both Rab7 and RNF167 signals were measured for each cell. CTL = control. Two-way ANOVA with Tukey’s post hoc test was used on the dataset collected from three independent experiments, totaling 38 to 54 cells corresponding to 190–270 vesicles analyzed. Errors bars (red) represent mean ± SEM, ns = non-significant, **, *p* = 0.0026, ****, *p* < 0.0001. (**E**) Manders’ overlap coefficient analysis between Rab7 and RNF167. Non-parametric one-way ANOVA Kruskal–Wallis test was used. Errors bars (red) represent mean ± SEM, *n* = 30–42 cells, ns = non-significant, ***, *p* = 0.0002.

## Data Availability

The data that support the findings of this study are available from the corresponding author upon reasonable request.

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
