# Peer review of "Membrane Targeting and GTPase Activity of Rab7 Are Required for Its Ubiquitination by RNF167"

_ijms, 2022, doi:10.3390/ijms23147847_

Round 1
Reviewer 1 Report
Molecular and cellular approaches were combined to highlight the functions of Rab7 and RNF167. The results and the discussion provide an integrated view of the molecular mechanisms involving Rab7 and RNF167. I found the manuscript written in an extremely clear manner. The purpose of each experiment is very well explained, and results seem solid to me.
Minor comments:
- Lines 460-461, a reference could be added to support this statement.
- The introduction should mention the organisms expressing Rab7 and RNF167, as well as the origin of the cells used in this study.
Author Response
Molecular and cellular approaches were combined to highlight the functions of Rab7 and RNF167. The results and the discussion provide an integrated view of the molecular mechanisms involving Rab7 and RNF167. I found the manuscript written in an extremely clear manner. The purpose of each experiment is very well explained, and the results seem solid to me.
We are pleased that the reviewer finds “the manuscript written in an extremely clear manner”. We are thankful that the reviewer took the time to read our manuscript and comment on our work.
Minor comments:
Concern 1. Lines 460-461, a reference could be added to support this statement.
Response 1. We have now referenced this statement in the text.
Concern 2. The introduction should mention the organisms expressing Rab7 and RNF167, as well as the origin of the cells used in this study.
Response 2. We have now included, in the introduction, some examples of the organisms expressing Rab7 and RNF167. We also added the cell type used during the study for the general results. In Materials and Methods, we specified the organism and the tissue of origin for each cell line.
Reviewer 2 Report
Authors describe identification of RNF167 as an ubiquitin ligase which ubiquitinates Rab7 GTPase and regulates its function. Further they document that Rab7 must be in GTP-bound state anchored to membranes for K191 and 194 ubiquitination and lysosomal localization. Finally, they show that RNF167-mediated ubiquitination of Rab7 variants identified in Charcot-Marie-Tooth type 2B patients is impaired. This manuscript represents complete piece of work that is important for better understanding of molecular pathogenesis of CMT type 2B disease.
Genes are expressed and proteins are produced or synthesized. Mutations are in genes and amino acid residue substitutions are in proteins. Other statements are laboratory jargon.
Rab7 is encoded by RAB7 or RAB7A gene. It should be stated in the text. Human genes are written in italics and proteins are not. Authors have to discriminate if they describe genes or proteins. This is not the same. Cells are transfected with DNA, not proteins.
Figure 2
B. Rather: Relative Memb/Cyto ratio of Rab7
Figure 3
B. Rather: Relative level of Lamp1
Figure 4
D. There is one significant difference in lysosome positioning in D, but in the text L269b is “lysosome positioning is not significantly affected. Clarify.
Figure 6
B. The difference for V162M is not convincing. Do you have statistical analysis?. In L356 is stated “was reduced”. Clarify this in the text.
Materials and Methods
Chapter 4.11 contains more than a sequence of DsiRNA. Title should be corrected. Silencing of RAB7 gene?
Chapter 4.11 disrupts the immunofluorescence methods. Should be moved after 4.1 where plasmid constructions are described. DNA methods go together.
Chapter 4.12 is continuation of 4.10. Could be a second paragraph of 4.10, without new number.
Other comments
L147, auto-ubiquitination but in the text which refers to D (L132) is: “results indicate that Rab7 ubiquitinbation,…. was observed when RNF167 and UBE2D1 activities where unaltered”. Rab7 is ubiquitinated and this is not a ligase, so it is not auto-ubiquitination. Should be corrected: ubiquitination.
L157, lysine-to-arginine substitution
L167, and the influence of GFP ubiquitination (GFP contains lysines)
L201, “Mutagenized lysines in Rab7” is not correct statement. Rather: lysine residue substitutions in Rab7
L202, with the respective plasmids encoding indicated proteins. Proteins are indicated in panel A.
L219, HeLa cells producing both Rab7 and RNF167
L232, in vivo
L252 with respective plasmids encoding indicated proteins
L283 Blot is representative result
L288, Rab7 level on RFN167L314, exogenous level of Rab7
L344, dependent
L379, Schematic representation of Rab7 domain structure and amino acid residues substituted in Charcot-Marie Tooth Type 2B variants
L381, amino acid residue substituted
L384, transfected with respective plasmids encoding indicated Rab7 protein variants
L398, DNA repair
L401, RNF167 controls such processes
L443, In the present study we show that
L454, substitution of amino acid residues
L457, amino acid residues
L458 This let us propose that
L464, substituted
L478, exogenous expression of a gene encoding RNF167 or increased level of RNF167
L482, membrane proteins
L491, 2xHA-hRab7 is a protein but the gene encoding this protein was synthesized and cloned
L500 Mutagenesis of a gene encoding hRab7
L550, 24h or 24 hours? Abbreviation 24h should be introduced.
L573, Laemmli 2x buffer
L614, producing cells
L640, The ubiquitination assay
L651, Laemmli 2x buffer
L657, separation of membranes . There is more than one membrane in the cell.
L665, pelleted membranes
L669, transferrin receptor
L670, tubulin
L682, Immunofluorescence microscopy
Author Response
ANSWERS TO REFEREES’ COMMENTS
Referee: 2
Authors describe identification of RNF167 as an ubiquitin ligase which ubiquitinates Rab7
GTPase and regulates its function. Further they document that Rab7 must be in GTP-bound
state anchored to membranes for K191 and 194 ubiquitination and lysosomal localization.
Finally, they show that RNF167-mediated ubiquitination of Rab7 variants identified in Charcot-
Marie-Tooth type 2B patients is impaired. This manuscript represents complete piece of work
that is important for better understanding of molecular pathogenesis of CMT type 2B disease.
We are pleased that the reviewer finds that our study “important for better understanding of
molecular pathogenesis of CMT type 2B disease”. We thank the reviewer for taking the time to
read and comment on our work.
Minor points:
Genes are expressed and proteins are produced or synthesized. Mutations are in genes and
amino acid residue substitutions are in proteins. Other statements are laboratory jargon.
Rab7 is encoded by RAB7 or RAB7A gene. It should be stated in the text. Human genes are
written in italics and proteins are not. Authors have to discriminate if they describe genes or
proteins. This is not the same. Cells are transfected with DNA, not proteins.
Thank you for your comment and we apologize for the apparent confusion our text may have
caused. The changes have been made in the text. We also indicated, in the introduction, that
we are studying Rab7A.
Figure 2
B. Rather: Relative Memb/Cyto ratio of Rab7
We have made the change to Figure 2.
Figure 3
B. Rather: Relative level of Lamp1
The changes have been done in Figures 4B and 5B.
Figure 4
D. There is one significant difference in lysosome positioning in D, but in the text L269b is
“lysosome positioning is not significantly affected. Clarify.
This observation puzzles us. Although we subject ourselves to rigorous procedures to enhance
reproducibility in our assays, we cannot fully explain why siRab7#1 + RNF167 is not significant.
Our best hypothesis for the discrepancy in our data is that the presented result is from non-
Gaussian biological samples. The data (siRab7#1 + RNF167) clearly presents some
abnormality in the distribution of individual events within the analyzed population. Although we tested for outliers (with Rout = 1%), we did not remove any data. Also, since RNF167 does not influence Lamp1 distribution for each siRNA condition, we believe that the appropriate
conclusion remains because RNF167 protein has no effect on Lamp1 distribution in our study.
Nevertheless, in the revised manuscript, we have changed the term “significantly” to
“conclusively”. The text now reads “[...] ratios of perinuclear to peripheral Lamp1-positive
vesicles show that lysosomes positioning is not convincingly affected in Rab7-depleted cells
when compared to control cells [...]”
Figure 6
B. The difference for V162M is not convincing. Do you have statistical analysis?. In L356 is
stated “was reduced”. Clarify this in the text.
This is an interesting question but we do not perform statistical analyses of ubiquitinated protein, at least not by Western blotting.
From our point of view, we do not believe that we can appropriately quantify Rab7 protein
ubiquitination due to inherent limitations with western blotting. One major issue is the absence
of a suitable control, on the membrane, for the immunoprecipitated Rab7 protein (i.e. Rab7~UB
vs unmodified Rab7 on the same membrane during a single WB). This is a real problem. In our
hands, the signal of stain-free signal from a membrane, which is normally used for
quantification, only allows us to detect the bands from the antibody used for
immunoprecipitation, not the immunoprecipitated Rab7 protein modified by ubiquitination. Also, in our experimental analysis setup, we separately blotted on different membranes for
ubiquitinated GFP-Rab7 (90% of IP GFP, WB anti-UB) and for evaluating the efficiency of Rab7
protein immunopurification (10 % of IP GFP, WB anti-GFP). In this context, which does not even
take into account the possible protein saturation of the blotting membranes, we do not consider that using the signal from an anti-GFP is an appropriate control for the anti-UB signal from different membranes.
Because we do not believe this method is appropriate for publication, we have not added the
proposed analysis for the ubiquitination of Rab7 in the presence of RNF167. However, as per
the methodology described below, we have quantified the result presented in Figure 6B of our
article and we determined that the level of ubiquitination of the CMT2B variants is indeed
reduced, being at least ~50% of the level of ubiquitination obtained with WT Rab7 protein.
Methodology: To analyze Rab7 ubiquitination, we made a relation between the stain-free signal
of the antibody band (signal from denatured anti-GFP used for IP) acquired from membranes
used for either anti-UB or anti-GFP blot. Using the immunoblot chemiluminescence signal
captured by the CCD camera and the stain-free signal from the membrane, we calculated the
ratio from the signal from the IP for WT or CMT2B Rab7 (either blotted anti-UB or anti-GFP)
over the stain-free signal of a specific membrane (membrane WB either UB or GFP). This
provides two ratios: (i) “anti-UB” and (ii) “anti-GFP”. Then, these ratios were used to obtain a relative ratio determined for each Rab7 protein and read (taking V162M as an example): V162M anti-UB ratio (i) / V162M anti-GFP ratio (ii). We normalized this V162M ratio to the one from WT, allowing us to graph as a percentage of WT Rab7 ubiquitination.
Round 2
Reviewer 2 Report
Presentation of results was very greatly improved. The manuscript is now suitable for publication, after correction of one mistake: RAB7A , not Rab7A, is a correctly written gene name. Correct in L50, to be as it is in L357 and 472.